# Research on the Formation and Plugging Risk of Gas Hydrate in a Deepwater Drilling Wellbore: A Case Study

Haodong Chen [1,*], Ming Luo [1], Donglei Jiang [1], Yanhui Wu [1], Chuanhua Ma [1], Xin Yu [1], Miao Wang [2], Yupeng Yang [1], Hexing Liu [1] and Yu Zhang [3]

1  Engineering Technology Operation Center of CNOOC (China) Co., Ltd., Hainan Branch, Haikou 570100, China
2  Engineering Technology Branch of CNOOC Energy Development Co., Ltd., Tianjin 300450, China
3  College of Petroleum Engineering, Yangtze University, Wuhan 430100, China
*  Correspondence: chenhaodong_cnooc@126.com

**Abstract:** At present, the formation mechanism of gas hydrate (hereinafter referred to as hydrate) plugging in the wellbore during deepwater drilling is not clear, so there are problems such as the overuse of hydrate inhibitors and the low utilization efficiency of inhibitors. Therefore, in view of the risk of hydrate formation and plugging under different working conditions during deepwater drilling, research was carried out on the wellbore hydrate formation area and wellbore hydrate deposition and plugging. Taking an atmospheric well in the South China Sea as an example, the wellbore annulus temperature field under different working conditions was combined with the hydrate formation phase curve to analyze the hydrate formation plugging risk under different working conditions during deepwater drilling, and the hydrate formation risk region of the wellbore under different working conditions was obtained. The effects of the drilling fluid circulation rate, injection temperature and drilling fluid viscosity in the wellbore annulus on the risk zone and subcooling of the wellbore hydrate formation were predicted. A deepwater drilling wellbore hydrate deposition plugging model was developed, based on which the dynamic deposition of the hydrate in the wellbore was predicted quantitatively. The results of the study showed that: (1) Increasing the circulation rate of drilling fluid, drilling fluid inlet temperature and drilling fluid viscosity during deepwater drilling can effectively reduce the hydrate formation region and subcooling, thus reducing the hydrate formation. (2) The risk of plugging by hydrate formation basically does not occur during normal drilling. (3) Under the condition of using seawater bentonite slurry drilling fluid, the safe operating time for stopping drilling is 20 h, and the safe operating time for shutting in and killing the well is 30 h.

**Keywords:** deepwater drilling; natural gas hydrate; formation risk; annulus temperature; plugging

## 1. Introduction

The exploration and development of deepwater oil and gas has been developing in recent years, and land drilling is gradually moving to deepwater and ultra-deepwater [1,2]. As the water depth increases, the risk of gas hydrate formation during drilling operations also increases. In deepwater and ultra-deepwater drilling operations, when natural gas intrusion or drilling encounters natural gas reservoirs, natural gas in the reservoir enters the wellbore and hydrates are easily formed due to the low temperature and high-pressure conditions in the wellbore. This brings new problems and challenges to the design of drilling and production process parameters, well control, spacer design, etc. [3–5]

The hydrate is a crystalline inclusion in which the molecular water cage captures a light hydrocarbon substance, such as methane or ethane. The hydrate is stable under a high pressure and low temperature, which is a challenge to ensure wellbore circulation during deepwater drilling. Hydrate crystals may clump or adhere to the wellbore during

deepwater drilling, resulting in complete wellbore annulus closure and serious operational and safety hazards associated with hydrate plugging repairs [6]. Therefore, improving the accuracy of prediction tools and models has become the focus of research, which is conducive to effectively reducing the risk of wellbore plugging during deepwater drilling.

In order to effectively control and reduce the risk of hydrate formation plugging, different research on hydrate formation plugging has been carried out by domestic and foreign scholars successively. The research on hydrate formation risk is mainly based on the theory of hydrate formation phase equilibrium; Van der Waals and Platteeuw [7] first developed a model for calculating hydrate formation conditions (Van der Waals-Platteeuw model) based on the chemical potential theory. Since then, Parrish and Pransnitz [8], John and Holder [9], Du Yahe and Guo Tianmin [10] and Sun et al. [11] have all carried out different degrees of hydrate formation models based on the Van der Waals-Platteeuw model. The research made it more accurate to predict hydrate formation under different conditions.

The research on hydrate plugging risks focuses on the theory and technology of hydrate formation plugging risks for deepwater gas well development and subsea multiphase pipeline transmission plugging risks. Both Wang Zhiyuan et al. [12–14] and Ren Guanlong et al. [15] studied the problem of insufficient hydrate deposition in the test column of deepwater gas wells and its impact degree on testing operations, and established a hydrate formation calculation model and deposition prediction model in the test column to study the hydrate formation plugging risk in the development of deepwater gas wells over formation on the basis of the regional prediction of hydrate formation risk in the test column of deepwater gas wells. Liu et al. [16] and Li Sangfang et al. [17] studied the formation process of hydrate plugging during the testing of deepwater gas wells and analyzed the influencing law of factors such as gas and liquid production and inhibitor concentration on hydrate plugging. Li et al. [18] experimentally studied the hydrate formation, aggregation and deposition characteristics under natural gas pipeline conditions and proposed a method to characterize hydrate plugging in pipelines. Sinquin et al. [19] and Aman et al. [20] carried out experimental studies on hydrate formation and deposition under annulus flow conditions in high-pressure horizontal loops and found that not all generated hydrates are deposited and attached to the pipe wall to form a hydrate deposit layer, and some of the hydrates are carried out of the loop by the gas. Gong Jing et al. [21], based on the established hydrate flow guarantee experimental platform combined with the hydrate power formation mechanism, multiphase flow law and reliability theory, carried out research on the theory and technology of multiphase pipeline transportation containing hydrates and its blockage risk. They effectively predicted the hydrate generation and blockage risk in a multiphase pipeline from both qualitative and quantitative aspects, which is helpful to ensure the safe operation of multiphase flow in the submarine transportation system.

At present, hydrate research focuses on hydrate formation mechanisms, hydrate formation plugging risks for deepwater gas well development and the theory and technology for subsea multiphase pipeline transportation plugging risks. However, there is still a lack of in-depth research on the risk of plugging by hydrate formation in the wellbore during deepwater drilling, and it is necessary to systematically conduct research on the formation mechanism and prevention methods of hydrate formation and plugging obstacles in deepwater drilling wellbores. The author mainly focuses on different working conditions such as drilling circulation, kill well, shut-in and stop drilling in the deepwater drilling process, and takes an atmospheric well in the South China Sea as an example of semi-anti-hydrate drilling fluids such as HEM drilling fluid and PLUS/KCL drilling fluid used in the field drilling process to reduce the risk of hydrate formation and plugging. Considering the economic benefits and drilling speed, etc., the use of seawater as a drilling fluid in some deepwater drilling in the South China Sea is expected to speed up the drilling speed and save costs, but the lack of a risk assessment of hydrate formation plugging in the deepwater drilling process restricts the replacement of drilling fluid. Therefore, it is necessary to study the hydrate formation and deposition characteristics under different

flow conditions in the wellbore during deepwater drilling, to establish a prediction model of hydrate flow obstacles under different working conditions during deepwater drilling and to reveal the formation mechanism of hydrate formation and plugging obstacles under different working conditions [22–25].

## 2. Regional Prediction of Wellbore Hydrate Formation Risk

Prediction of the wellbore temperature field during deepwater drilling is difficult compared to onshore and shallow water drilling because the temperature gradient of seawater is opposite to the temperature gradient of the formation and there is a complex heat exchange between seawater and the water trap. The low temperature of the seafloor affects the density and rheology of drilling fluids, while hydrates may form due to the combined effect of the low temperature and high pressure on the seafloor, so to accurately predict the formation of hydrates in the wellbore during deepwater drilling, it is necessary to first understand the temperature distribution in the wellbore [26].

The temperature distribution during deepwater drilling is characterized by three stages: ① From the bottom of the well to the seabed location, the wellbore temperature field decreases slowly with the decrease in formation temperature. ② Due to the low temperature of the deepwater seabed and the serious influence of seawater temperature field disturbance, the test fluid between the bulkhead and the tubing has poor thermal insulation and rapid heat dissipation, resulting in a rapid decline in wellbore temperature at locations near the mudline. ③ The seawater section from the seabed to the wellhead location up to 1000 m or more will warm the oil and gas slowly and the wellhead temperature will not increase significantly with test production changes [2,27,28].

### 2.1. Hydrate Formation Risk Prediction Model

2.1.1. Wellbore Temperature Field Equation

An accurate prediction of the wellbore temperature field is the basis for determining the hydrate formation area during deepwater drilling. In the process of deepwater drilling, the flow and heat transfer of the drilling fluid in the wellbore annulus is partially simplified considering the complexity of the flow and heat transfer, assuming the following [29]:

(1) The drilling fluid flow in the wellbore annulus is a steady-state flow, and the drilling fluid physical parameters and flow heat transfer parameters are the same in the annulus cross-section.

(2) Only radial heat transfer is considered in the heat transfer process, not longitudinal heat transfer, and the formation heat transfer is unsteady considering its time effect and steady-state heat transfer within the wellbore.

(3) The influence of rock cuttings on the heat capacity and thermal conductivity of drilling fluid is ignored, and the heat generated by the viscous dissipation of drilling fluid is ignored.

Assuming that the fluid flows through a stationary control body unit, the transient heat transfer mechanism when a heat and mass transfer occurs simultaneously in the fluid in the annulus is modeled as follows; the mass conservation equation for the drilling fluid in the control body unit is as follows.

$$\frac{\partial(\rho_l)}{\partial_t} + \nabla \times (\rho_l v) = 0 \tag{1}$$

The momentum conservation equation for the drilling fluid in the control body unit:

$$\frac{\partial(\rho_l v)}{\partial_t} + \nabla \times (\rho_l v v) = -\nabla \times (p) + \nabla \times (F_{vis}) + \rho_l g \sin\theta \tag{2}$$

The energy conservation equation for the drilling fluid in the control body unit:

$$\frac{\partial}{\partial t}[\rho(U_l + \frac{1}{2}|v|^2)] + \nabla \times [\rho(U_l + \frac{1}{2}|v|^2 v)] = \nabla \times (Q_l) + \nabla \times (F_{vis}) - \nabla \times (pv) + \rho_l v g \sin\theta \tag{3}$$

In the equation: $\rho_l$–density of drilling fluid, kg /m$^3$; $v$–flow rate, m/s; $t$–time, s; $F_{vis}$–viscous force, Pa; $g$–free-fall acceleration, m/s$^2$; $\theta$–well angle, rad.

### 2.1.2. Seawater Temperature Field Equation

The environment surrounding the wellbore includes seawater and a stratigraphic area. The temperature distribution in the seawater region along the vertical depth is very complex and is influenced by many factors, including latitude, currents, season and depth. The seawater temperature at different depths can be expressed using the Levitus model [30].

$$
\begin{aligned}
T_w &= [T_{surf}(200 - z) + 13.68z]/200, \; z < 200 \\
T_w &= a_2 + (a_1 - a_2)/(1 + \exp(z - a_0)/a), \; 200 \leq z \leq 4000
\end{aligned}
\tag{4}
$$

In the equation: $a_0$ = −137.137; $a_1$ = 39.39839; $a_2$ = 2.30713; $a_3$ = 402.73177; $T_w$–seawater temperature, °C; $T_{surf}$–seawater surface temperature, °C; $z$–well depth, m.

### 2.1.3. Thermodynamic Equations for Hydrate Formation

The thermodynamic equations for natural gas hydrate formation can be obtained from the thermodynamic equilibrium theory based on the equilibrium relationship between the water phase, the gas phase and the degree of hydrate lattice systematization in the crystal structure of natural gas hydrates [31].

$$
\frac{\Delta\mu_0}{RT_0} - \int_{T_0}^{T_H} \frac{\Delta H_0 + \Delta C_K(T_H - T_0)}{RT_{H^2}} dT_H + \int_{P_0}^{P_H} \frac{\Delta V}{RT_H} dp_H = \ln(f_w/f_{wr}) - \sum_{i=1}^{l} M_i \ln(1 - \sum_{j=1}^{L} \theta_{ij})
\tag{5}
$$

$$
\ln(f_w/f_{wr}) - \sum_{i=1}^{l} M_i \ln(1 - \sum_{j=1}^{L} \theta_{ij})
\tag{6}
$$

In the equation: $\Delta\mu_0$–chemical potential difference between empty hydrate lattice and water in pure water at the standard state, J/mol; $R$–universal gas constant, J/(mol·K); $T_0$–temperature in the standard state, K; $T_H$–phase temperature of hydrate formation, K; $\Delta H_0$–specific enthalpy difference between empty hydrate lattice and pure water, J/kg; $\Delta C_k$–difference in specific heat capacity of empty hydrate lattice and pure water, J/(kg ·K); $p_H$–the phase pressure of hydrate formation, Pa; $p_0$–pressure at standard condition, Pa; $\Delta V$–specific volume difference between empty hydrate lattice and pure water, m$^3$/ kg; $f_w$–fugacity of water in the water-rich phase, Pa; $f_{wr}$–Fugacity of water at reference state $T_H$ and $p_H$, Pa; $M_i$–the number of type $i$ pores per unit water molecule in the hydrate phase; $L$– number of components that can generate hydrates; $\theta_{ij}$–occupancy of type $j$ guest molecules in type $i$ pores, f; $x_w$–molar fraction of water in the water-rich phase, dimensionless; $y_w$–the activity coefficient of water in the water-rich phase, f.

### 2.1.4. System of Equations Fixed Solution Conditions

(1)　Boundary conditions for the set of wellbore temperature field equations during fugitive gas intrusion under drilling conditions.

$$
\begin{aligned}
p(0, t) &= p_s \\
q_g(D, t) &= q_g \\
q_c(D, t) &= 0
\end{aligned}
\tag{7}
$$

(2)　Boundary conditions for the set of equations under pressure well conditions.

$$
\begin{aligned}
p(D, t) &= p_B + p_q \\
q_g(D, t) &= 0 \\
q_c(D, t) &= 0
\end{aligned}
\tag{8}
$$

(3) The fluid temperature at the entrance of the drill column can be measured directly, and the boundary conditions of the temperature field.

$$T_P(0,t) = T_{in} \tag{9}$$

(4) The temperature of the fluid in the drill column and the fluid in the annulus are equal at the bottom of the well.

$$T_P(D,t) = T_a(D,t) \tag{10}$$

In the equation: $p_s$–atmospheric pressure, Pa; $D$–well depth, m; $q_g$–intrusion of gas, kg/s; $q_c$–generation of rock chips, kg/s; $p_B$–stratigraphic pressure, Pa; $T_p$–temperature inside the drill column, K; $T_{in}$–temperature of the drilling fluid at the injection point of the drilling column, K; $T_p$–temperature inside the drill column, K; $T_a$–Temperature in the annulus, K.

2.1.5. Discretization and Solution of the Theoretical Model

The direct analytical solution of the wellbore temperature field equation is difficult, and the multi-temperature gradient environment of deepwater drilling requires a discrete solution for the equation, so a numerical algorithm is needed to find the discrete solution for the above equation.

The specific solution procedure is as follows.

(1) Preliminary assumptions about the pressure at node $y$ at time $n + 1$;
(2) The temperature of the annulus at $n + 1$ at node $y$ is solved separately from the temperature field equation;
(3) If a natural gas hydrate can be formed, then any node $y$ in this loop is a point in the hydrate formation region;
(4) Substitute the determined parameters into the energy equation and solve for the pressure at node $y$ at time $n + 1$. If the newly calculated pressure at node y at $n + 1$ time is within the error tolerance, stop the calculation of node y and use the parameter calculated at node $y$ as the known condition for the calculation of $y + 1$ point, otherwise return to step (1) and re-estimate until the condition holds.

*2.2. Generate Risk Region Predictions*

In order to verify the algorithm of this paper, one hypothetical deepwater normal pressure well was used for the simulation. The assumed parameters of the deepwater atmospheric well are: the drill bit is 3000 m below the surface, the water depth is 1800 m, the 889 mm casing is down to 2500 m below the surface, the outer diameter of the drill column is 127 mm, the diameter of the drill bit is 311 mm, the inner diameter of the throttle line is 76.2 mm, the density of the drilling fluid is 1.05 g/cm$^3$, the apparent viscosity is 1 to 16 mPa·s, the dynamic shear stress is 3 MPa, the flow rate during normal drilling is 4000 L/min during normal drilling, 1200 L/min during kill well, 300 L/min during shut-in, 0 L/min during stop drilling, a formation pressure coefficient of 1.0, subsea temperature of 276 K and geothermal gradient of 4 °C/100 m after entering the formation and a formation rupture equivalent drilling fluid density of 1.2 g/cm$^3$. The theoretical model for predicting the gas hydrate formation area in deepwater drilled wellbores is solved based on the previous solution method by applying the base data, and the results are discussed below.

(1) Prediction of gas hydrate formation region under different drilling fluid circulation rates.

The hydrate phase curve is combined with the wellbore temperature field at different drilling circulation rates to convert the temperature–pressure point at the hydrate phase equilibrium to the temperature–depth point at wellbore conditions. Comparing the temperature–depth curve in the wellbore with the hydrate phase curve, the hydrate formation region in the wellbore can be obtained. When the hydrate phase curve is to

the right of the wellbore temperature curve, the area surrounded by the two curves is the hydrate formation region. The larger the length of this region in the longitudinal direction, the larger the hydrate formation region; if the width is larger in the lateral direction, the larger the hydrate formation supercooling [32], the easier the hydrate formation and the faster the hydrate formation rate.

In order to save drilling costs, a large number of sites use seawater as a drilling fluid with a drilling fluid viscosity of 1 mPa·s. The risk of gas hydrate formation in the wellbore at different drilling fluid circulation rates during deepwater drilling is shown in Figure 1.

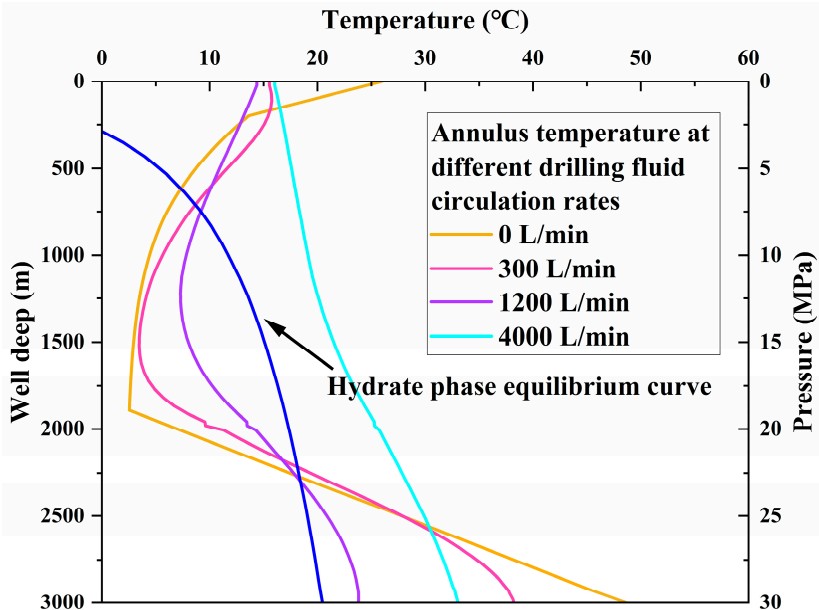

**Figure 1.** Risk of gas hydrate formation in the wellbore at different drilling fluid circulation rates.

As shown in the figure, as the flow rate decreases, the hydrate formation area gradually increases, and the subcooling degree gradually increases, so the natural gas hydrate is more likely to be formed. Therefore, increasing the drilling fluid flow rate during the drilling process will help to prevent hydrate formation.

(2)  Prediction of gas hydrate formation region under different drilling fluid inlet temperatures.

In the deepwater drilling process, hydrates are not formed during the normal drilling circulation, and the time of shut-in and stop drilling is generally short, so the circulation flow rate of 1200 L/min is taken as the kill well condition. The risk of gas hydrate formation at different drilling fluid injection temperatures is shown in Figure 2.

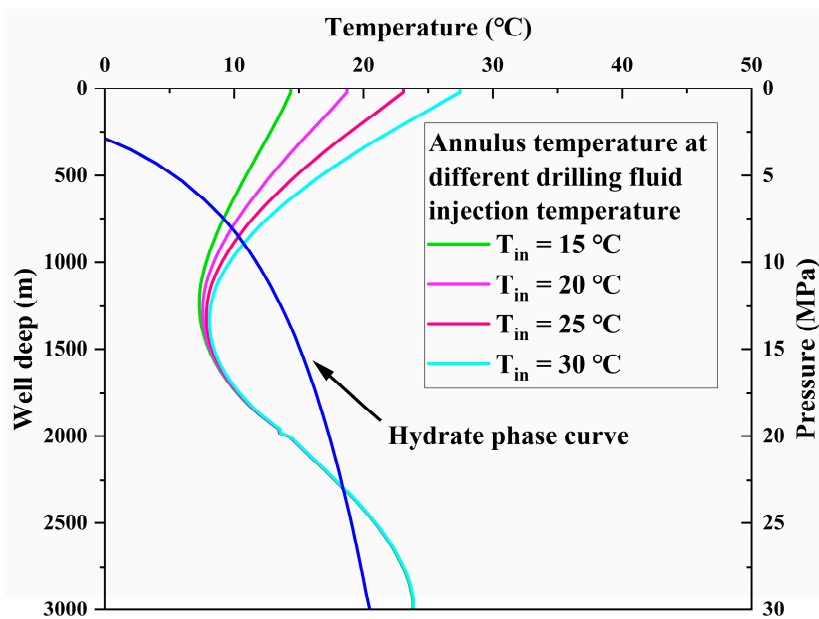

**Figure 2.** Risk of gas hydrate formation under different drilling fluid inlet temperatures.

As shown in the figure, the inlet temperature of drilling fluid can significantly change the temperature distribution in the annulus under the circulation condition, and the area of the natural gas hydrate formation gradually decreases with the increase of the inlet temperature of the drilling fluid, and the subcooling degree gradually decreases. Therefore, it is suggested that the inlet temperature of the drilling fluid can be increased to prevent the formation of hydrates by heating the drilling fluid returned from the well.

(3)　Prediction of gas hydrate formation region under different drilling fluid viscosity.

The risk of gas hydrate formation under different drilling fluid viscosity is shown in Figure 3, taking the pressurized well condition with circulation rate of 1200 L/min.

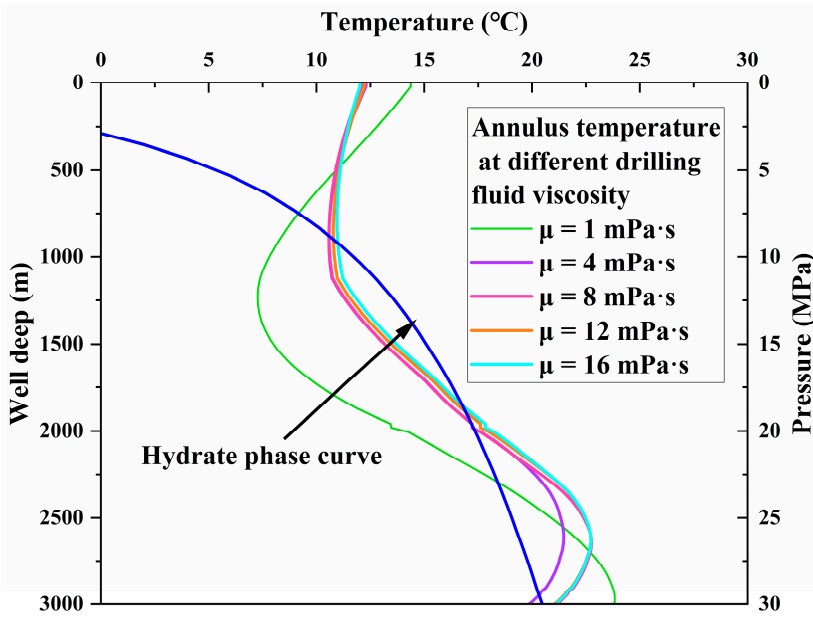

**Figure 3.** Risk of gas hydrate formation under different drilling fluid viscosity.

As can be seen from the figure, the change in drilling fluid viscosity has a significant effect on the viscous force $F_{vis}$ in the wellbore. As the viscosity of the drilling fluid increases,

the area of the gas hydrate formation gradually decreases, and the subcooling degree gradually decreases. When the drilling fluid viscosity is greater than 8 mPa·s, the effect of viscosity on the wellbore annulus temperature decreases. Therefore, in deepwater drilling, the drilling fluid viscosity should be increased to prevent the formation of hydrates.

## 3. Hydrate Deposition Plugging Model

### 3.1. Modeling of Deposition Plugging

Deepwater drilling is usually dominated by the liquid phase. The gas phase at the center of the tubing flow is dispersed in the liquid phase and flows at a high velocity along with the liquid phase. A schematic diagram of the hydrate deposition and plugging in the wellbore is shown in Figure 4.

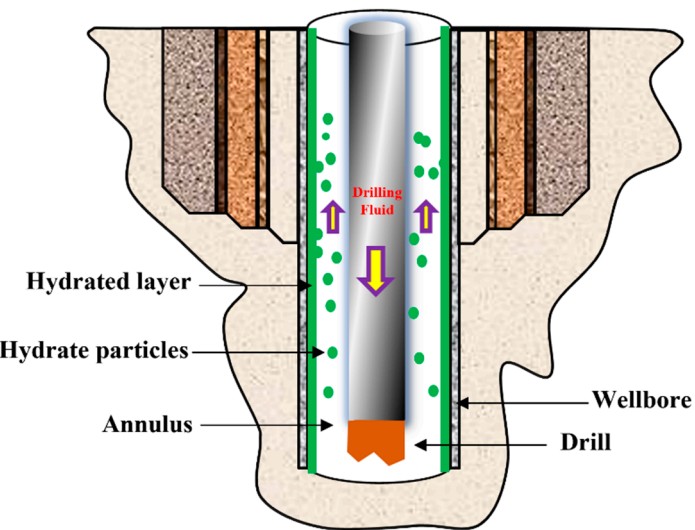

**Figure 4.** Formation and deposition of hydrates in the wellbore.

During the drilling circulation, when the temperature and pressure meet the conditions of hydrate formation in a certain location in the wellbore, hydrates will be formed, and that location is the hydrate formation region. The formed hydrate will be carried by the drilling fluid, and part of the hydrate will be deposited on the tubing wall to form a hydrate deposit layer, resulting in a reduced effective overflow region and increased pressure drop in the wellbore. With the increasing thickness of the hydrate layer, the wellbore is gradually blocked, which is the reason for the flow obstruction caused by the hydrate.

In this paper, based on the results of hydrate loop experiment research carried out by Lorenzo et al. [33] and Turner et al. [34], the following formula for calculating the rate of hydrate formation in the wellbore is constructed:

$$R_{hf} = \frac{\mu A_s k_1 M_h}{M_g} \exp(-\frac{k_2}{T_f})(\Delta T_{sub}) \tag{11}$$

In the equation: $R_{hf}$–rate of hydrate formation in the wellbore, kg/(s·m); $\mu$–a parameter characterizing the effect of mass and heat transfer processes on the rate of hydrate formation, dimensionless; $A_s$–air–liquid contact area, m$^2$; $k_1$–intrinsic kinetic parameters, 2.608 × 10$^{16}$ kg·m$^{-2}$·K·s$^{-1}$; $T_f$–temperature of the fluid in the wellbore, K; $\Delta T_{sub}$–subcooling the difference between the hydrate formation temperature and the fluid temperature, which is the driving force for hydrate formation), K.

Some of the hydrate formed in the wellbore is transported with the drilling fluid, and some of the hydrate is deposited and attached to the tubing wall. In the wellbore, hydrates formed in the tubular wall will adhere to the tubular wall due to the strong adhesive force from the inner wall of the tubular column [35]. The hydrates formed at the gas entrained by

the drilling fluid will be transported for a longer distance due to the high-speed carrying effect of the drilling fluid. Compared to the smaller extent of hydrate formation in the tubular column of subsea tubulars, the deposition of hydrate particles formed at the drilling fluid entrained gas on the tubular wall can be neglected, so the deposition of the hydrate formed at the tubular wall on the tubular wall is the main cause of tubular column plugging. The hydrate deposition rate is calculated by the following equation.

$$R_{hd} = \frac{A_f r_f k_1 M_h}{M_g} \exp(-\frac{k_2}{T_f})(\Delta T_{sub}) \tag{12}$$

In the equation: $R_{hd}$–rate of hydrate deposition, kg/(s·m); $A_f$–area of the interface between the liquid surface of the tube wall and the gas, m²; $r_f$–shrinking pipe diameter with hydrate deposition (effective pipe diameter), m.

### 3.2. Prediction Method

Hydrate formation is a relatively slow process and most of the hydrate formed will be carried by the high-speed flowing drilling fluid, so even if the temperature and pressure in some areas of the tubing column meet the conditions for hydrate formation, a plug will not form immediately. The hydrate deposition plugging model is used to predict the distribution of the hydrate deposition layer thickness, and then appropriate control measures can be taken to ensure that no plugging occurs in the wellbore. The steps are as follows:

(1) Calculate the temperature and pressure field of the wellbore and determine the hydrate formation region of the wellbore by combining the phase equilibrium conditions of the hydrate formation. The area where the wellbore temperature is lower than the hydrate formation phase equilibrium temperature is the hydrate formation area.
(2) Calculate the hydrate formation rate and deposition rate according to the hydrate formation rate equation and deposition rate equation.
(3) From the calculated hydrate formation rate and deposition rate, the growth of the hydrate layer thickness at different depths on the pipe wall with time can be obtained to know the hydrate plugging condition in the pipe column.

## 4. Case Study

Hydrate deposition in an atmospheric pressure well in the South China Sea under different drilling conditions is taken as an example. The parameters of the well are: the drill bit is 3039 m below the surface, the water depth is 1893 m, the 889 mm casing is down to 2535 m below the surface, the outer diameter of the drill column is 127 mm, the diameter of the drill bit is 311 mm, the inner diameter of the throttle line is 76.2 mm, the density of the drilling fluid is 1.05 g/cm³, the density of the seawater bentonite slurry drilling fluid is 16 mPa·s, the dynamic shear stress is 3 MPa, the flow rate is 4000 L/min during normal drilling, 1200 L/min during kill well, 300 L/min during shut-in, 0 L/min during stop drilling, a formation pressure coefficient of 1.0; a subsea temperature of 276 K and a geothermal gradient of 4.61 °C/100 m after entering the formation and a formation rupture equivalent drilling fluid density of 1.18 g/cm³.

A theoretical model for predicting the gas hydrate formation area in deepwater drilled wellbores is solved by applying the underlying data. The hydrate formation region of an atmospheric well in the South China Sea with different drilling conditions is shown in Figure 5.

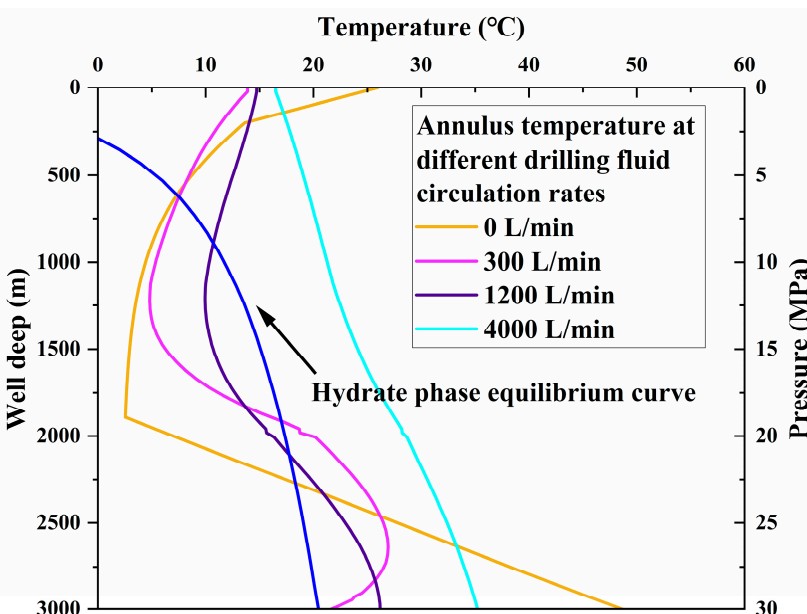

**Figure 5.** Risk of gas hydrate formation in the case wellbore at different drilling fluid circulation rates.

The corresponding circulation rates for the different working conditions of the well are shown in Table 1.

**Table 1.** Circulation rate at different working conditions.

| Drilling Conditions | Rate of Flow (L/min) |
| --- | --- |
| Drilling circulation | 4000 |
| Kill well | 1200 |
| Shut-in | 300 |
| Stop drilling | 0 |

From Figure 5, it can be seen that when the circulation rate in the wellbore is 0 L/min, 300 L/min and 1200 L/min, there is a hydrate formation risk region in the wellbore, which means that hydrate deposition will gradually start to occur in the wellbore and a hydrate layer with a certain thickness will gradually appear on the inner wall of the wellbore. The hydrate deposition in the wellbore under kill well, shut-in and stop drilling conditions was calculated to obtain the hydrate layer formation in the inner wall of the wellbore at different times and different rates of flow. The variation of the hydration layer thickness with time under the stop drilling condition is shown in Figure 6, the variation of the hydration layer thickness with time under the shut-in condition is shown in Figure 7 and the variation of the hydration layer thickness with time under the kill well condition is shown in Figure 8.

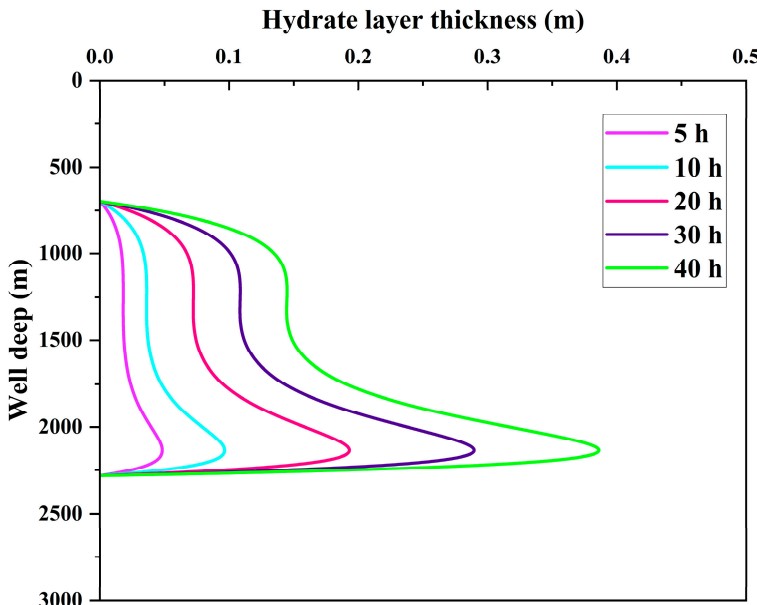

**Figure 6.** Variation of hydrated layer thickness with time under stop drilling condition.

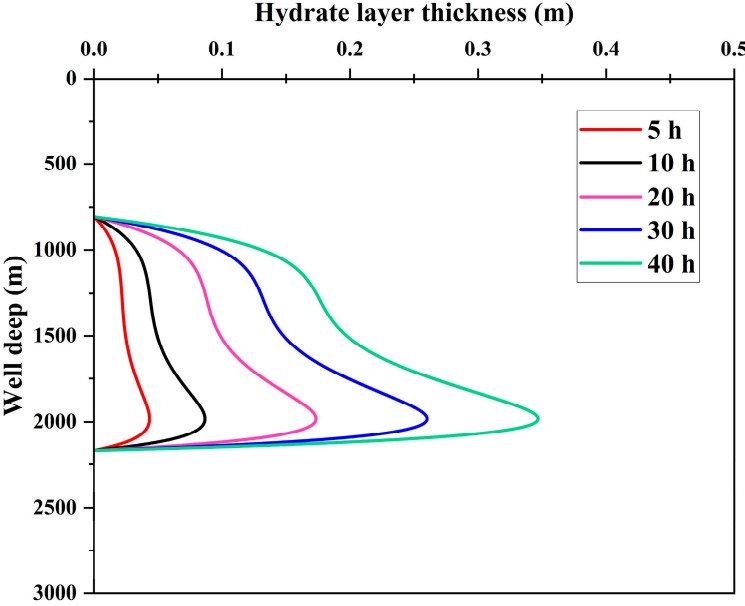

**Figure 7.** Variation of hydrated layer thickness with time under shut-in condition.

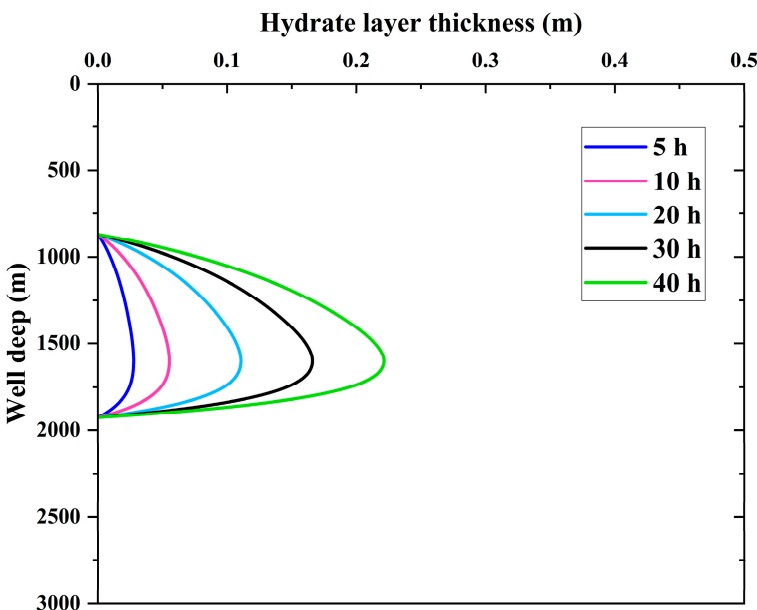

**Figure 8.** Variation of hydrated layer thickness with time under kill well condition.

It can be seen from the above figure that under a certain flow rate, the thickness of the wellbore hydrate layer gradually increases with time, mainly because hydrate deposition is a gradual accumulation process. With the continuous deposition of the hydrate, the hydrate layer on the pipe wall gradually thickens, and the effective inner diameter of the wellbore gradually decreases. Wellbore hydrate deposition is strongly influenced by the drilling fluid circulation rate, and the lower the drilling fluid circulation rate, the greater the fluctuation of hydrate layer distribution on the tubing wall. The well depth structure of an atmospheric well in the South China Sea is shown in Figure 9. The internal diameters of the different types of casing are shown in Table 2.

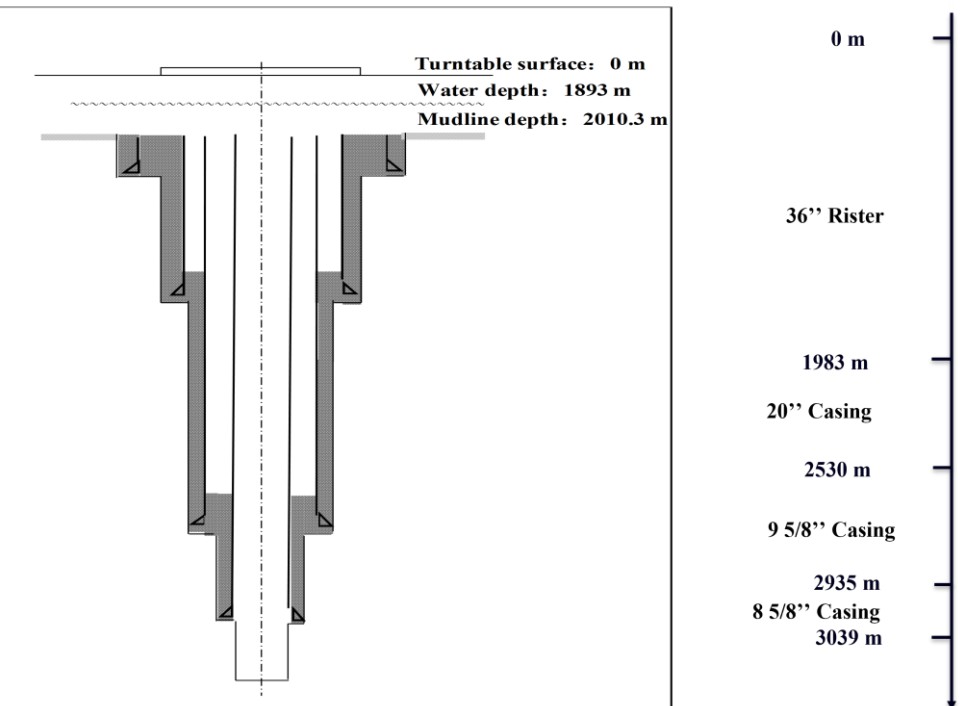

**Figure 9.** Structure of a normal pressure well body in the South China Sea.

**Table 2.** Inner diameter of casing for different models.

| Casing Model | Inner Diameter/m |
|---|---|
| 36″ | 0.889 |
| 20″ | 0.476 |
| 13–3/8″ | 0.32 |
| 9–5/8″ | 0.22 |
| 8–5/8″ | 0.201 |

At a depth of 1000 m to 2000 m, the internal diameter of the casing with 20″ is 0.476 m; therefore, no hydrate plugging will occur under the stop drilling, shut-in and kill well conditions at this depth with seawater bentonite slurry as the drilling fluid for 40 h, and no hydrate will occur under the drilling conditions with a 4000 L/min circulation flow. Therefore, no hydrate plugging will occur within 40 h of drilling with seawater bentonite slurry as the drilling fluid at this depth. When drilling to 2789 m to 2954 m, using a 13–3/8″ casing with an inner diameter of 0.32 m, no hydrate plugging will occur within 30 h of the stop drilling, shut-in and kill well conditions with seawater as the drilling fluid at this depth. When drilling to 2954 m to the target layer, using a 9–3/8″ casing with an internal diameter of 0.22 m, no hydrate plugging will occur under the stop drilling, shut-in and kill well conditions at this depth for 20 h with seawater as the drilling fluid.

According to the actual drilling conditions of an atmospheric well in the South China Sea, seawater bentonite slurry was used as the drilling fluid, and there was no risk of hydrate blockage during the normal drilling cycle. When drilling to 2865 m, drilling was stopped for about 30 h due to special well conditions. When drilling again, hydrate plugging occurs, the torque increases and the drilling fluid does not circulate. After 50% thermodynamic inhibitor ethylene glycol is added for about 50 min, the hydrate blockage is removed and normal drilling is resumed. The actual drilling conditions on site show that the models of hydrate formation and plugging in a deepwater drilling wellbore and the quantitative prediction model of wellbore hydrate dynamic depositions established in this paper are basically consistent with the risk of hydrate formation and plugging in deepwater drilling sites, which has good on-site guidance significance.

## 5. Conclusions

(1) The wellbore temperature field of a deepwater drilling wellbore is obtained according to the wellbore temperature field equation, and the wellbore hydrate formation risk region under different drilling conditions is established by combining the deepwater drilling wellbore annulus temperature field and hydrate formation phase equilibrium curve.

(2) The effects of the drilling fluid circulation rate, injection temperature and drilling fluid viscosity in the wellbore annulus on the risk zone and the subcooling of wellbore hydrate generation were predicted. Increasing the drilling fluid circulation rate, drilling fluid inlet temperature and drilling fluid viscosity during deepwater drilling can effectively reduce the hydrate formation zone and subcooling, thus reducing the hydrate formation.

(3) Based on the hydrate deposition plugging model, the variation of the hydrate layer with time was quantitatively predicted under different working conditions. Under the condition of using seawater bentonite slurry drilling fluid, the safe operation time for stopping drilling is 20 h, and the safe operation time for shut-in and kill well is 30 h, which provides a safe operation time for on-site drilling construction.

**Author Contributions:** Conceptualization, H.C. and M.L.; methodology, H.C.; software, D.J.; validation, C.M., X.Y. and M.W.; formal analysis, H.C.; investigation, Y.Y.; resources, Y.Z. and H.L.; data curation, Y.W.; writing–original draft preparation, H.C.; writing–review and editing, M.L.; visualization, Y.W.; supervision, D.J. All authors have read and agreed to the published version of the manuscript.

**Funding:** This research received no external funding.

**Data Availability Statement:** Not applicable.

**Acknowledgments:** This research was supported by the Deepwater project team of CNOOC Hainan Branch and the offshore deepwater Oil and Gas Development Institute of Yangtze University. During the research process, the project also received the support of postgraduates such as Pu Lei, Zhou Shanshan, Jiang Qisheng and Liu Qinglin. Here, we would like to express our heartfelt thanks to the research institutions and individuals who have provided us with support and help.

**Conflicts of Interest:** The authors declare no conflict of interest.

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
