# Peer review of "Research on the Formation and Plugging Risk of Gas Hydrate in a Deepwater Drilling Wellbore: A Case Study"

_processes, doi:10.3390/pr11020488_

Round 1

Reviewer 1 Report

The paper discusses and analyzes the risk of formation of gas hydrate and the plugging risk in deepwater drilling wellbore. I find the paper interesting from a scientific point of view but I suggest some improvements before publication.

1- Check the typos and please reduce the number of repetitions in the sentences.
2- The discussion of results of the case study can be improved.
3- I was wondering if the authors have thought about a validation of their models. 

Reviewer 2 Report

Interesting case study on the formation and plugging risk of gas hydrate in deepwater drilling wellbore. The manuscript discusses the results from the hydrate formation and plugging model for deepwater drilling wellbore. The model, methodology, and results provided are rational. However, some minor errors need to be corrected before this manuscript can become suitable for publication. My questions/suggestions are as below.

My questions/suggestions

1.     Cross-referencing for figures is wrong. Please recheck all figure numbers in the manuscript.

2.     Some typos in the manuscript:

a.     In the Title, ‘wellbor’ should be ‘wellbore’

b.     Line 212, ‘Parameters of the well.’ looks like an abrupt sentence.

c.      In the acknowledgments, line 401, it should be ‘During’ instead of ‘uring’.

3.     Why is figure 1 repeated again as figure 4 in the text? Authors can refer to figure 1 if they want to discuss the results from it.
